# Development of 50 InDel-based barcode system for genetic identification of tartary buckwheat resources

**Hwang-Bae Sohn[1]⊚, Su-Jeong Kim[1]⊚, Su-Young Hong[1], Sin-Gi Park[2], Dong-Ha Oh[3], Sunghoon Lee[4], Hwa Yeun Nam[1], Jung Hwan Nam[1], Yul-Ho Kim[1]***

**1** Highland Agriculture Research Institute, National Institute of Crop Science, Pyeongchang, Gangwon-do, Republic of Korea, **2** TheragenEtex Bio Institute, TherageneEtex Inc., Suwon, Gyeonggi-do, Republic of Korea, **3** Department of Biological Science, Louisiana State University, Baton Rouge, LA, United States of America, **4** EONE-DIAGNOMICS Genome Center Co. Ltd., Incheon, Republic of Korea

⊚ These authors contributed equally to this work.
* kimyuh77@korea.kr

**Data Availability Statement:** All relevant data are within the paper and its Supporting Information files.

## Abstract

Tartary buckwheat (*Fagopyrum tataricum* Gartn.) is a highly functional crop that is poised to be the target of many future breeding efforts. The reliable *ex situ* conservation of various genetic resources is essential for the modern breeding of tartary buckwheat varieties. We developed PCR-based co-dominant insertion/deletion (InDel) markers to discriminate tartary buckwheat genetic resources. First, we obtained the whole genome from 26 accessions across a superscaffold-scale reference genome of 569.37 Mb for tartary buckwheat cv. "Daegwan 3–7." Next, 171,926 homogeneous and 53,755 heterogeneous InDels were detected by comparing 26 accessions with the "Daegwan 3–7" reference sequence. Of these, 100 candidate InDels ranging from 5–20 bp in length were chosen for validation, and 50 of them revealed polymorphisms between the 26 accessions and "Daegwan 3–7." The validated InDels were further tested through the assessment of their likelihood to give rise to a single or a few PCR products in 50 other accessions, covering most tartary buckwheat genome types. The major allele frequencies ranged from 0.5616 at the TB42 locus to 0.9863 at the TB48 locus, with the average PIC value of 0.1532 with a range of 0.0267–0.3712. To create a user-friendly system, the homology of the genotypes between and among the accessions were visualized in both one- (1D) and two-dimensional (2D) barcode types by comparing amplicon polymorphisms with the reference variety, "Daegwan 3–7." A phylogenetic tree and population structure of the 76 accessions according to amplicon polymorphisms for the 50 InDel markers corresponded to those using non-synonymous single nucleotide polymorphism variants, indicating that the barcode system based on the 50 InDels was a useful tool to improve the reliability of identification of tartary buckwheat accessions in the germplasm stocks.

**Funding:** This study was financially supported by grants from Co-operative Research Program for Agriculture Science and Technology Development funded by the Rural Development Administration (RDA), Republic of Korea to YHK (Project No. PJ01189401) and TheragenEtex Inc. provided support for this study in the form of salaries for SGK. The specific role of this authors is articulated in the 'author contributions' section. The RDA evaluated the project every year for 5 years (2016~2020) and had a role in decision to submit for publication, but the funders had no additional role in study design, data collection and analysis, or preparation of the manuscript.

**Competing interests:** The authors have read the journal's policy and the authors of this manuscript have the following competing interests: SGK is a paid employee of TheragenEtex Inc. and SL serves on the board of directors of EONE DIAGNOMICS Genome Center Co. Ltd. This does not alter our adherence to PLOS ONE policies on sharing data and materials. There are no patents, products in development or marketed products to declare.

## Introduction

The eudicot family Polygonaceae consists of 27 reported species, including the 2 cultivated species, common buckwheat (*Fagopyrum esculentum* Moench) and tartary buckwheat (*F. tataricum* Gaert.) [1,2]. Buckwheats (*F. esculentum* and *F. tataricum*) are cultivated, as short-season pseudo-cereals, in many countries and regions, including China, Nepal, Russia, Europe, Korea, and Japan [3,4]. Following domestication in upland southwestern China, a multitude of tartary buckwheat landraces were created by ancient farmers through both artificial and natural selection [5]. After cultivation in China, these landraces were first dispersed to Nepal, Russia, Europe, Korea, and Japan [5–9]. Several characteristics of tartary buckwheat, such as its self-pollinating nature and ability to adapt to various regions (ranging from high altitude zones to lowland), led to both morphology and genetically divergent pockets of landraces [10–13]. These landraces have been largely replaced by fewer landraces with specific desired characteristics such as high-yield, drought resistance, and flavor [5,14].

Tartary buckwheat is a major food crop in high altitude zones (e.g., high mountain areas of southern China and the Himalayan hills) due to its frost tolerance under poor soil conditions. Tartary buckwheat has also been used more recently for a variety of purposes, ranging from various transitional food products to medical use [15,16]. The grain of tartary buckwheat is considered to be an ideal functional food source for humans because it is richer in proteins, fats, vitamins, rutin, quercetin, and other flavonoids than common buckwheat [17–23]. The popularity of tartary buckwheat has been steadily increasing, and its cultivation has broadened into areas far outside its original region [10,11,24,25]. This is a testament to tartary buckwheat's adaptability, which will be crucial in Earth's changing climate [14]. Breeding for commercial varieties of tartary buckwheat improves agricultural traits, such as lodging, late maturity, and low yield [26,27]; however, this cultivation resulted from a narrow genetic base from a few varieties only, which have produced modern varieties that are not well adapted to a variety of growing environment. The narrow genetic diversity within a commercial variety is due to the rigid quality required by farmers and processors, the finite use of exotic germplasm, restricted breeding strategies, and individual plant selection [28–30]. Thus, more genetic variation among tartary buckwheat genotypes could be used to improve a specific trait of interest due to the narrow genetic diversity of modern commercial varieties [14,31,32].

The identification of tartary buckwheat germplasms is vital to the enhancement of genetic diversity. Researchers have been focused on developing molecular markers using germplasms and morphological descriptors to provide high discrimination power. There are many methods for variety discrimination based on DNA polymorphisms, such as random amplification of polymorphic DNA (RAPD), amplified fragment length polymorphism (AFLP), inter-simple sequence repeat (ISSR), simple sequence repeats (SSRs), and sing nucleotide polymorphism (SNP). Insertion-deletion polymorphisms (InDels) are gaining more attention among molecular breeding scientists because they are easy to use, co-dominant (fully informative), and relatively abundant [33–40]. In tartary buckwheat, SSR [9], AFLP [41], RAPD [42,43], ISSR [44] and SNP [45,46] have been used to assess genetic variation, but no study has used a variety of discrimination methods with InDel markers due to the difficulty of finding DNA polymorphisms in tartary buckwheat accessions.

Until recently, a lack of large-scale genome sequencing information made the introduction of InDels as genetic markers in non-model species (e.g., common and tartary buckwheat) difficult. In recent years, published studies have provided the draft genomes of common buckwheat [47], and a tartary buckwheat cultivar "Pinku1" [48] in the family Polygonaceae. More importantly, a wealth of polymorphism data can now be obtained from a massive amount of sequencing data using high-throughput sequencing technologies [49]. In addition, Sohn et al.

[38] reported that the InDel-based barcode system focuses on usability, and provided an efficient resource management system. In addition to the InDel-based barcode system, morphological descriptors are now emerging that aim to identify tartary buckwheat accessions, which can be used to verify key genomic factors that explain or predict major agronomic traits.

We present a high-quality draft genome assembly and annotation for the tartary buckwheat. *F. tataricum* cv. "Daegwan 3–7." Using comparative genomics approaches, we revealed 171,926 homogeneous InDels and 53,755 heterogeneous InDels found in 26 tartary buckwheat accessions, which we compared to the draft genomes of the tartary buckwheat cultivar "Daegwan 3–7." Among them, 50 polymorphic InDels from the 26 accessions were selected by gel electrophoresis, which were converted to barcodes by comparing amplicon polymorphisms with the reference sequence. We also revealed the geographical distribution of 73 accessions through population structure analysis using the barcodes. For the genetic identification of tartary buckwheat resources, we made a user-friendly barcode system based on 50 InDel-based genotypes of 73 accessions. As a user-friendly system, the homology between the accessions can be predicted in both one (1D) and two-dimensions (2D) as blocks. Our platform could be used in genetic research and breeding programs, as well as for efficient resource management systems for tartary buckwheat.

## Materials and methods

### Plant materials

The accessions of 73 tartary buckwheat plants (S1 Table) were provided by the National Agrobiodiversity Center (Jeonju, Republic of Korea) and the Highland Agriculture Research Institute (Pyeongchang, Republic of Korea). The accessions were planted in a greenhouse located at the Highland Agriculture Research Institute (37°68′N and 128°73′E, 779 m altitude), Gangwon Province, Republic of Korea. These were self-pollinated 3 times by the single seed descent (SSD) method by growing the individual plants in flowerpots (10cm in diameter) [50]. We grew an additional four generations of HLB1001 using the SSD method, which resulted in the selection of a progeny, "Daegwan 3–7" for a draft genome of tartary buckwheat. A total of 26 accessions were collected (China, India, Nepal, Bhutan, Pakistan, and USA).

To investigate the biodiversity and comparative relationships between the accessions, collected accessions were sown in 2017 and grown in the greenhouse at the Highland Agriculture Research Institute. We separated the plants by developmental stage: the vegetative growth stage (1–40 days after planting, DAP), flowering stage (40–80 DAP), and yield stage (80–90 DAP). At the yield stage, the color of the seed coat turned black, indicating that the seeds were completely maturated [51].

### Measurement of rutin and quercetin content

The yield stage seeds were used to measure their rutin and quercetin content. First, the mature seeds from the 76 accessions were milled into a fine powder, then the samples were immediately flash-frozen in liquid nitrogen and stored at −80°C until further use. A total of 0.10 g dry powder for each sample was mixed with 1 ml methanol, then was extracted at 80°C for 1 hr in a conventional Soxhlet apparatus. The extract was filtered through a 0.20 μm syringe filter (PTFE 13 mm, PALL Life Sciences, Ann Arbor, MI) for rutin content [23]. Ultra-performance liquid chromatography (Waters Corporation, Milford, MA, USA) was performed on an Acquity 1-class using a C18 column (2.1 mm × 100 mm, 1.7 μm) (Waters Corporation, USA) at 30°C. The mobile phase (1% formic acid in water/0.1% formic acid in acetonitrile) flowed by the gradient elution method (S2 Table) at a flow rate of 0.25 ml/min with a total injection volume of 10 μ*l*. Rutin quantity was estimated based on the linear calibration curve of standard

rutin and quercetin (Extrasynthese, France) under a detection wavelength of 259 nm. Three independent sample analyses were performed for each sample. All statistical analyses were performed using ver. 3.6.1 of the R software.

## DNA sequencing and *de novo* assembly

After 21 DAP, the leaves in the third node of the progeny "Daegwan 3–7" were used to create a draft genome. First, 3-week-old leaves from the bulk of 10 plants were collected for genomic DNA extraction using a standard CTAB (cetyl trimethylammonium bromide) protocol [52,53]. Both short read (Illumina) and long read (PacBio) libraries were prepared according to the manufacturer's instructions for the entire genome of *F. tataricum*. A short insert paired-end (PE) library was prepared using a Illumina TruSeq Nano DNA Sample Preparation Kit (Illumina Inc., San Diego, Ca, USA) and 1~3 $\mu$g of DNA. A library profile analysis was performed with an Agilent 2100 Bioanalyzer (Agilent Technologies, USA) and qPCR quantification, then the libraries were sequenced to $2 \times 100$ bp on the Illumina HiSeq 2500 platform (Illumina, USA) from the TheragenEtex Bio Institute (TheragenEtex Inc., Suwon, Republic of Korea). PacBio SMRTbell libraries (2 kb, 5 kb, 10 kb, and 15 kb inserts) were prepared with the standard PacBio library preparation protocols, which are available at http://pacificbiosciences.com/. The sequencing was conducted on a PacBio RS II (Pacific Biosciences, USA) system using C4 chemistry and 240 min movies by the TheragenEtex Bio Institute.

The short and long reads were assembled separately. The de novo assembly of the PacBio long reads was performed using Fast Alignment and CONsensus [54] with default parameters. The short reads were assembled with SOAPdenovo2 [55] with default parameters. The initial contigs were merged using HaploMerger2 [56]. Both short and long reads were then used to construct scaffolds with SSPACE [57], which led to the superscafold-scale draft genome of tartary buckwheat cv. "Daegwan 3–7." The N50 length of the final genome assemblies was evaluated.

## Identification and validation of the polymorphic InDels

Genomic DNA was extracted for whole-genome resequencing from the 26 accessions that had self-pollinated 3 times using the SSD method. Three-week-old leaves from the bulk of 10 plants were collected for genomic DNA extraction using a standard CTAB protocol [52,53]. The data consisted of 101-bp reads, which were generated using the HiSeq 2500 sequencer. The Short Oligonucleotide Alignment Program 2 (SOAP2) was used to map the raw pair-end reads onto the reference genome (tartary buckwheat cv. "Daegwan 3–7"). For each accession, more than 93% of the reads were properly aligned to the reference genome. Insert size was estimated by mapping the reads to the reference genome using the Burrows-Wheeler Aligner algorithm (bwa) [58] ver 0.5.9. The aligned reads were realigned at InDel positions using the GATK InDelRealinger algorithm [59] enhance the mapping quality. The base quality scores were recalibrated using the GATK TableRecalibration algorithm. The primer pair used to amplify each of the InDels were selected by the Primer3 software (http://primer3.sourceforge.net). We selected InDels with more than 10 bp and designed the primers accordingly to match the characteristics of each InDel using the Primer 3 software.

## Amplicon polymorphism assay and bar-coding process

PCR analysis was performed using 10 $\mu l$ reaction mixtures containing 20 ng of total genomic DNA, 2 pM of primer, and 5 $\mu l$ of GoTaq Green Master Mix (Promega, Madison, WI, USA). The PCR was performed at 95˚C for 5 min, followed by 35 subsequent rounds at 94˚C for 30 sec, 45˚C for 30 sec and 72˚C for 30 sec, using a Mastercycler pro 384 (Eppendorf, Germany).

The PCR products were separated by electrophoresis in 3% gels of certified low range ultra-agarose (Bio-rad) followed by LoadingSTAR staining (Dynebio, Republic of Korea). For the limitation of the number of candidates, we chose primer pairs that amplified PCR products that were 150–231 bp long. The allelic diversity of the InDels with the e-PCR products in 26 tartary buckwheat genomes were assessed via PIC (Polymorphism Information Content), which was defined as PIC $i = 1 - \sum_{j=1}^{n} p_{ij}^2$, where p $_{ij}$ is the frequency of the $j$th pattern for the $i$th marker [60,61].

The 50 InDel markers were selected according to their genotyping success and PCR band size. The discriminating power of the selected 50 InDel set for tartary buckwheat identification was evaluated using the 76 tartary buckwheat accessions. Primers were designed by targeting the InDel region in such a way that the genotypes of tartary buckwheat accessions would produce the same or different (insertion or deletion) amplicons relative to the reference genome, "Daegwan 3–7." Based on the results of InDel marker amplification, the same result with the reference genome was represented by an "a" while other results were represented by a "b." For bar-coding representation of the results, "a" and "b" were converted to white and black, respectively. The homology of the accessions were calculated after PCR amplification of all 50 InDels.

### Phylogenetic analysis and population structure among 73 tartary buckwheat accessions

The discriminating power of the selected 50 InDel set for tartary buckwheat identification were evaluated using the 76 tartary buckwheat accessions. The analysis was carried out with POWERMARKER Ver. 3.23 (Liu and Muse 2005; http//www.poxermarker.net). A phylogenetic tree for the 73 accessions was drawn based on the genotypes defined with the 50 InDels using the weighted neighbor-joining method with simple matching coefficients in the DARwin software [55]. To delineate clusters of individuals on the basis of their genotypes, the model-based STRUCTURE 2.3.4 [63] was used with the admixture model, a burn-in period of 100,000 iterations, and data from the 50 InDel loci. The number of clusters (k) was set to five, as this number maximized the Δk (ad hoc criterion) parameter. We used simulations with k values ranging from 2 to 10 with 5 replications, to calculate the LnP(D) value. The optimal k value can be chosen based on the maximum LnP(D) that provides a distinct population structure. On this basis, we chose K = 3 (three ancestral groups) for the tartary buckwheat populations. To check the quality of these analyses, a phylogenetic tree and a population structure for the 26 re-sequenced accessions were drawn based on the genotypes defined using the 6,622 non-synonymous SNPs.

## Results

### Genome assembly and whole-genome sequencing of tartary buckwheat accessions

We grew seven generations of tartary buckwheat cv. "Daegwan" using the SSD method and an isolated individual (designated as "Daegwan 3–7") to exclude any possible heterogeneity in the original seed lot. We produced a total of 43.83 and 32.17 Gb sequences from Illumina PE and Single-Molecule Real-Time (SMRT) sequencing platforms, respectively, which corresponded to 78x (Illumina PE, S2 Table) and 58x (SMRT, S3 Table) coverages (S1 Fig) for the reference genome. Combining the sequencing strategies of PacBio and Illumina Hiseq2500, we obtained a final draft assembly of 569.37 Mb in 2,886 scaffolds with 50% of the total sequence captured in 156 scaffolds larger than 886,968 bps (Table 1, N50).

**Table 1. Summary of *Fagopyrum tataricum* draft genome assembly process.**

| Steps | Type | No. of Sequences | Total bases | Longest sequence | N50 sequence | N90 sequence |
|---|---|---|---|---|---|---|
| Illumina assembly | Contigs | 37,711 | 416,936,361 | 318,774 | 29,297 | 5,269 |
| PacBio assembly | Contigs | 2,231 | 450,944,578 | 5,290,013 | 650,667 | 148,793 |
| Hybrid assembly | Contigs | 4,433 | 565,095,288 | 5,290,013 | 463,432 | 36,547 |
| Scaffolding / Gap-filling | Scaffolds | 2,886 | 569,372,063 | 5,875,542 | 798,933 | 61,611 |

We selected 26 tartary buckwheat accessions for whole-genome resequencing. These accessions originated or were popularized in different Asian and international regions, including 15 landraces from China, 4 from Nepal, 4 from India, 1 from the USA, 1 from Bhutan and 1 from Pakistan (Table 2). After 3 additional generations were grown using the SSD method, a panel of the 26 genotypes showed a broad extent of rutin content and phenotype variation (Figs 1 and 2). The 26 accessions showed a various distribution of rutin content compared to those of the 73 accessions (Fig 2). In particular, HLB1013 had the highest rutin content (1,921 mg/100g·DW), while HLB1006 had the lowest (1,250 mg/100g·DW) among the 26 accessions. Large differences were found in seed shape and seed coat color among the 26 accessions (Fig 1). Seed information conformed to the known the genetic variation of the 26 accessions.

**Table 2. Genome coverage and mean depth after the whole genome resequencing of 26 accessions.**

| Genotype | IT Number [a] | Country of origin | Genome coverage (%) [b] | Mean depth |
|---|---|---|---|---|
| HLB 1001 | IT 224676 | USA | 94.80 | 24.28 |
| HL1B 002 | IT 225083 | China | 93.64 | 26.60 |
| HLB 1003 | IT 225084 | China | 94.00 | 23.25 |
| HLB 1004 | IT225085 | India | 93.63 | 23.78 |
| HLB 1005 | IT225086 | India | 93.36 | 26.40 |
| HLB 1006 | IT225087 | India | 94.53 | 26.20 |
| HLB 1007 | IT225088 | Nepal | 92.99 | 24.58 |
| HLB 1008 | IT225089 | Nepal | 93.47 | 24.28 |
| HLB 1009 | IT225090 | Nepal | 93.21 | 27.30 |
| HLB 1010 | IT226673 | China | 93.64 | 24.68 |
| HLB 1011 | IT226674 | China | 94.22 | 25.89 |
| HLB 1012 | IT226675 | China | 94.57 | 25.79 |
| HLB 1013 | IT226676 | China | 93.92 | 24.44 |
| HLB 1014 | IT226677 | China | 94.06 | 25.74 |
| HLB 1015 | IT226678 | China | 94.19 | 25.53 |
| HLB 1016 | IT226679 | China | 94.01 | 24.93 |
| HLB 1017 | IT226680 | China | 94.18 | 25.66 |
| HLB 1018 | IT226682 | Nepal | 93.14 | 24.69 |
| HLB 1019 | IT261917 | China | 94.23 | 25.13 |
| HLB 1020 | IT261922 | China | 94.89 | 23.70 |
| HLB 1021 | IT261924 | Bhutan | 93.70 | 25.21 |
| HLB 1022 | IT278316 | China | 94.35 | 24.44 |
| HLB 1023 | IT278317 | China | 94.05 | 26.99 |
| HLB 1024 | IT278318 | China | 94.56 | 23.90 |
| HLB 1025 | IT278319 | Pakistan | 94.60 | 25.85 |
| HLB 1026 | IT226681 | India | 93.98 | 26.65 |

[a] National registration number managed by Nation Agrodiversity Center in Republic of Korea.

[b] Genome coverage was calculated based on a 569 Mbp genome size and 101 bp paired-end Illumina reads.

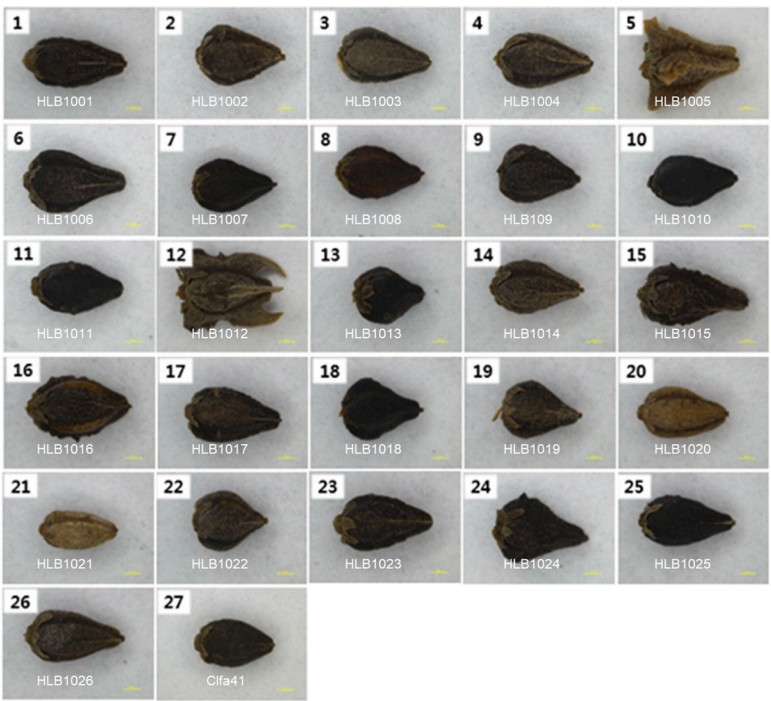

**Fig 1. The seed morphology of 26 re-sequenced tartary buckwheat accessions and Clfa41.**

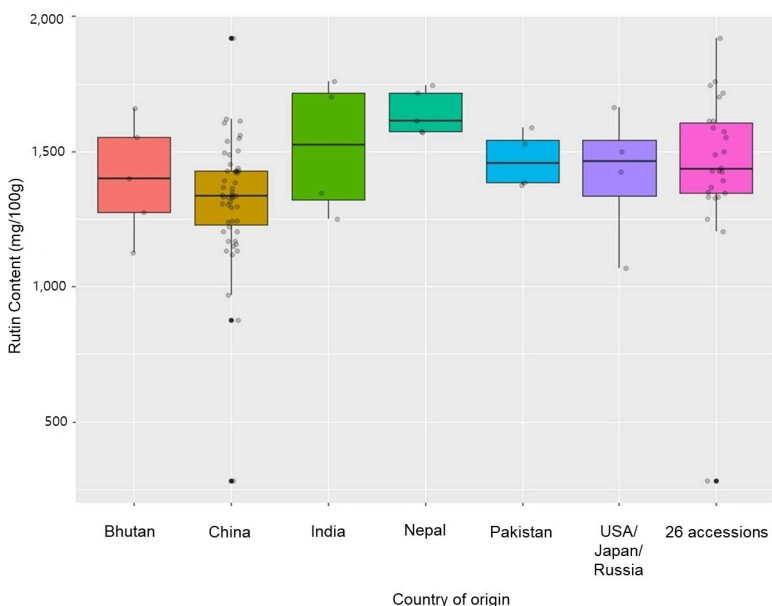

**Fig 2. Rutin content of 73 tartary buckwheat accessions according to country of origin.** The phenotypic segregation is shown in box-plot format. The interquartile region, median, and range are indicated by the box, the bold horizontal line, and the vertical line respectively. USA/Japan/Russia includes 4 accessions; 1 from the USA, 1 from Japan, and 2 from Russia. We selected 26 accessions for whole-genome resequencing that originated or were popularized in different Asian and international regions. For expanded details of the 73 tartary buckwheat accessions, see S1 Table.

We produced paired-end DNA reads at 23–27-fold depths (Table 2) for the 26 accession genomes and mapped them to the reference genome ("Daegwan 3–7"). The sequencing qualities of all the samples were high: 93%–95% of the accession sample reads were mapped to the reference genome, and 96%–97% of the reference genome was covered (S3 Table). The genetic variations (SNPs and small InDels) among the 26 accessions collected from the six countries were individually identified by Samtools and the Genome Analysis Tollkit GATK (see Methods). To increase the accuracy of the prediction, only the variations predicted by both methods were used for subsequent analysis. Bioinformatic analysis revealed 171,926 and 53,755 homogeneous and heterogeneous InDels, respectively (Fig 3). The heterozygous rates of all accessions were around 10%, reflecting the possibility of cross-pollination although self-fertilization was used (S4 Table).

## Selection of 50 InDels and constructing barcode-type database of 73 accessions

Sequence comparison at the nucleotide level of the tested genomes with the reference genome revealed InDels that could be easily used to identify tartary buckwheat resources in common laboratories. Consistent with what has been found in other organisms [38,64], the great majority of homogeneous InDels from all accessions were short with a dominance of 4-bp or fewer events (Fig 3). From this polymorphism data, we chose InDels with more than 10-bp for further analysis to help us identify them by PCR and gel-electrophoresis. We tested 100 primer sets, focusing on their ability to amplify PCR products in the 26 genome types, since this study aimed to establish PCR-based markers applicable for all tartary buckwheat accessions. PCR primers were designed to generate amplicons of varying sizes within the 150–231 bp interval (S5 Table). Among them, 50 InDel loci were widely distributed on whole chromosomes in the reference genome ("Pinku 1") reported previously (Fig 4). We selected the 50 InDels to validate the prediction accuracy and found that 98.2% of the PCR results in the 26 accessions were

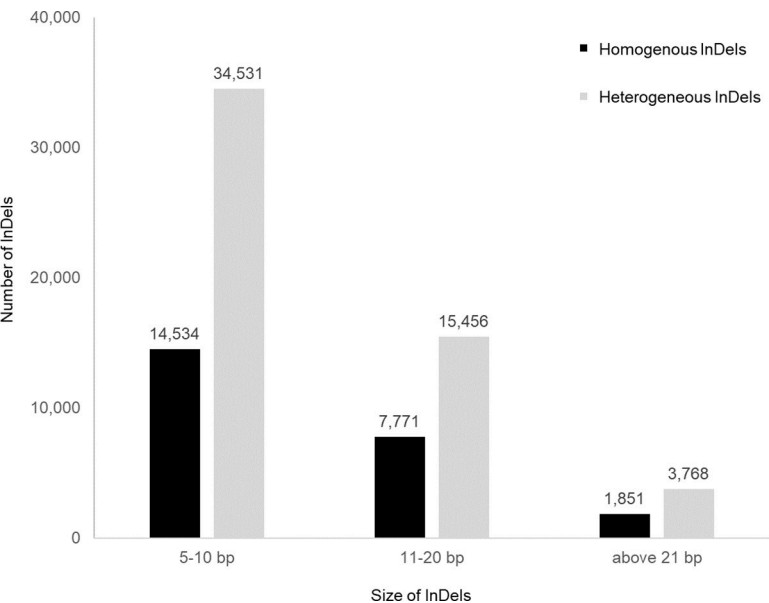

**Fig 3. Number of InDels in 26 tartary buckwheat accessions aligned with reference genome, "Daegwan 3–7."** The number of variants homogeneous to alternative genotype; both alleles are same to alternative genotype. The number of variants heterogeneous genotype; one allele is same to reference, and another to alternative genotype.

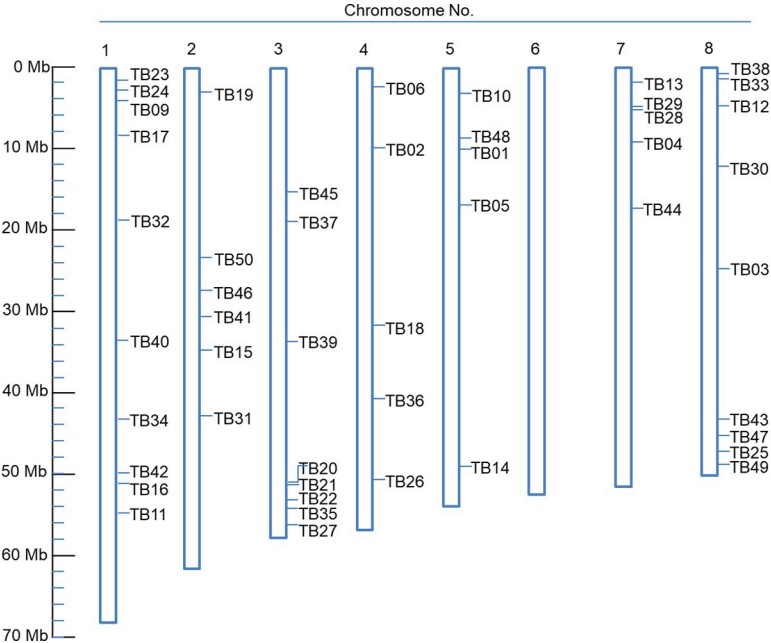

**Fig 4. Schematic of the chromosomal distribution of the tartary buckwheat InDels.** The number of the chromosome is shown on each chromosome of a reference genome of "Pinku1."

consistent with the predicted results. The results of the InDel analyses were highly accurate (S5 and S6 Tables). This high accuracy provided a solid foundation for our data analyses and provides high-quality data for future data mining.

Primers were designed by targeting the InDel region, in such a way that the genotypes of the 26 accessions would produce the same or different (insertion or deletion) amplicons relative to the reference genome (Fig 5). By using the previously reported barcode method [38,39], identical results to the reference genome "Daegwan 3–7" were represented by an "a," while the other results were represented by "b," which were presented as "white" and "black" barcodes, respectively. Fig 5 shows the barcode system applied to the tartary buckwheat accessions HLB1007 and HLB1008, which were converted to the standard 1D and widely-used 2D barcode types by comparing their amplicon polymorphisms to the reference genome, "Daegwan 3–7." This result highlights that the barcode types can be effectively used to investigate the degree of marker difference in tartary buckwheat accessions (Fig 6). The creation, stability and quality of the barcode system for genetic identification using the selected 50 InDels is thoroughly evaluated below.

By building a database of the 50 InDel polymorphisms for the 73 tartary buckwheat accessions (S2 Fig), we have established a more stable foundation for utilizing the barcode system to provide a promising tool for identifying the genetic resources of tartary buckwheat. Allele frequencies and average PIC value of the 50 InDel loci are shown in Table 1. The major allele frequencies ranged from 0.5616 at the TB42 locus to 0.9863 at the TB48 locus. The average PIC value was 0.1532, with a range of 0.0267–0.3712, which implied that the selected InDels could be applied for investigating polymorphisms of tartary buckwheat accessions (S7 Table). The genotypes of the 73 tartary buckwheat accessions were constructed with a two-dimensional barcode, and genotypic discrimination was analyzed by comparing it with the accessions of the barcode system. The average difference between the analyzed accessions was 7 InDels from a total of 50 InDels, while 6 accessions had differences in 12 InDels compared to the reference

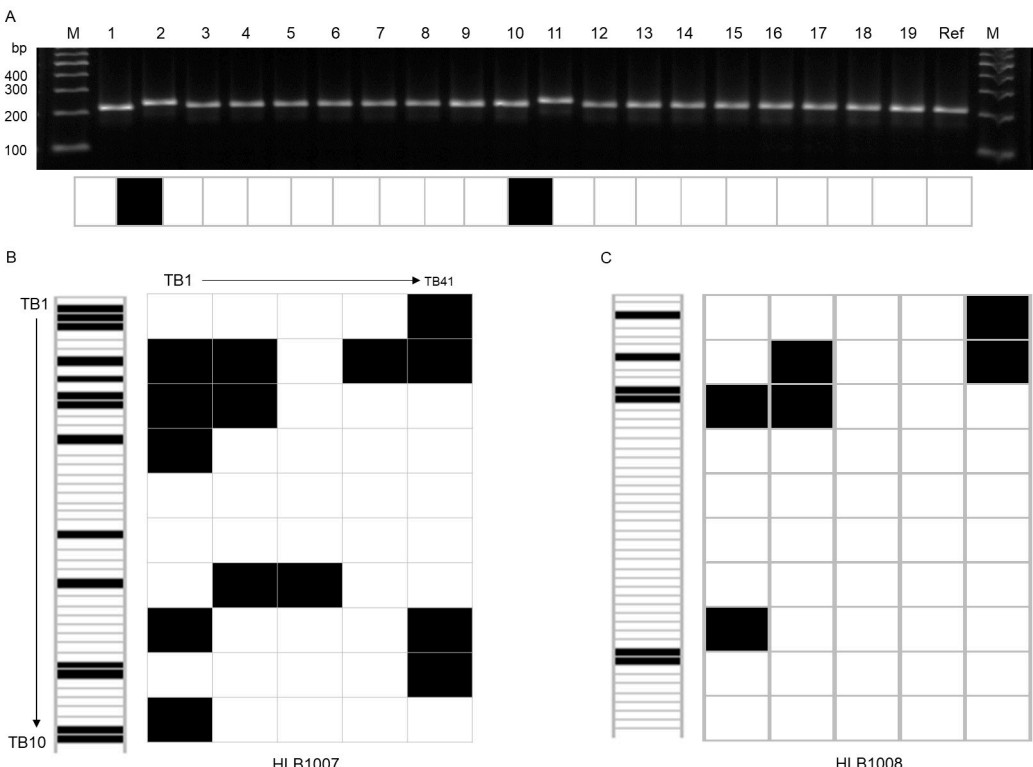

**Fig 5. Application of tartary barcode system to tartary accession, HLB1007 and HLB1008.** (A) Amplification products and barcode representation of the polymorphisms using InDel marker TB11 on tartary buckwheat accessions. PCR results using the 50 InDels in (B) HLB1007 and (C) HLB1008 were converted to standard 1D and widely-used 2D barcode types in comparison with those in "Daegwan 3–7." Identical results to the reference genome were represented by white, while other results were represented by black.

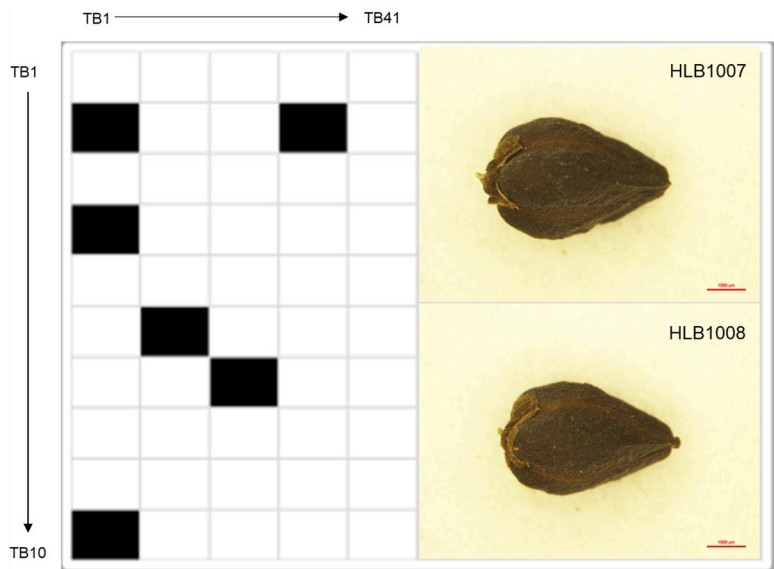

**Fig 6. Comparison of 2-dimensional barcode patterns between HLB1007 and HLB1008.** Identical results are represented by white, while the other results are represented by black. Seed morphology of HLB1007 and HLB1008.

genome, "Daegwan 3–7." By comparing the two-dimensional barcodes of the accessions, different blocks could be visually distinguished with ease, so that the genotype differences of the accessions could be determined (Fig 6). The discrimination power of genetic resources was high enough to broaden our understanding of both phylogenetic signals and population-level variation.

## Phylogenetic analysis and population structure in tartary buckwheat accessions

Phylogenetic analysis of the 6,622 non-synonymous SNP variants and population structure subdivided the panel into 2 primary populations (Fig 7): the reference group and the non-reference group. A similar result was attained with the SSR data, which indicated two separate group clusters of tartary buckwheat [9]. The 23 accessions had 4 subgroups within the non-reference group, and were classified according to their collection area. The phylogeny resolved subgroup 2 (3 Nepal, 1 Bhutan, and 2 China) as closer to subgroup 1 (8 China and 1 India) than subgroup 3 (3 India, 1 Pakistan, and 1 Nepal).

To validate the reliability of the 50 InDel markers for discriminating between tartary buckwheat accessions, bin maps were constructed for 73 tartary buckwheat accessions using the InDels (S6 Table). We analyzed the population structures of the studied accessions and reference groups. The result are shown in Fig 8. At K = 3, the clusters were anchored to Southeast Asia and China. Among subgroup 1, the accessions in Southeast Asia were almost entirely composed of the red component, while the Chinese accessions were composed of the red, green, and blue components. Subgroup 1 mainly consisted of accessions from Southeast Asia

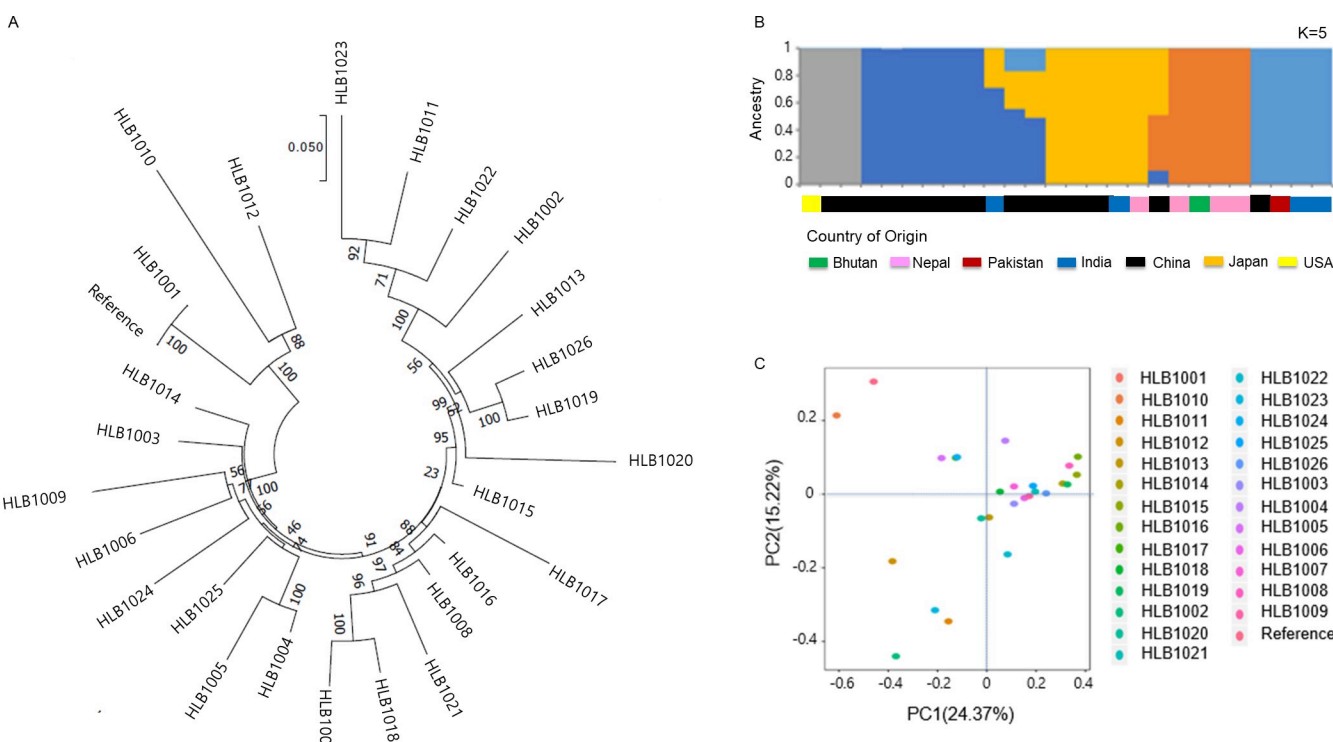

**Fig 7. Diversity analysis of 26 re-sequenced tartary buckwheat accessions.** (A) Phylogenetic analysis based on non-synonymous SNPs. (B) STRUCTURE [63] classification and (C) principal component analysis were similar. Each color represents country of origin.

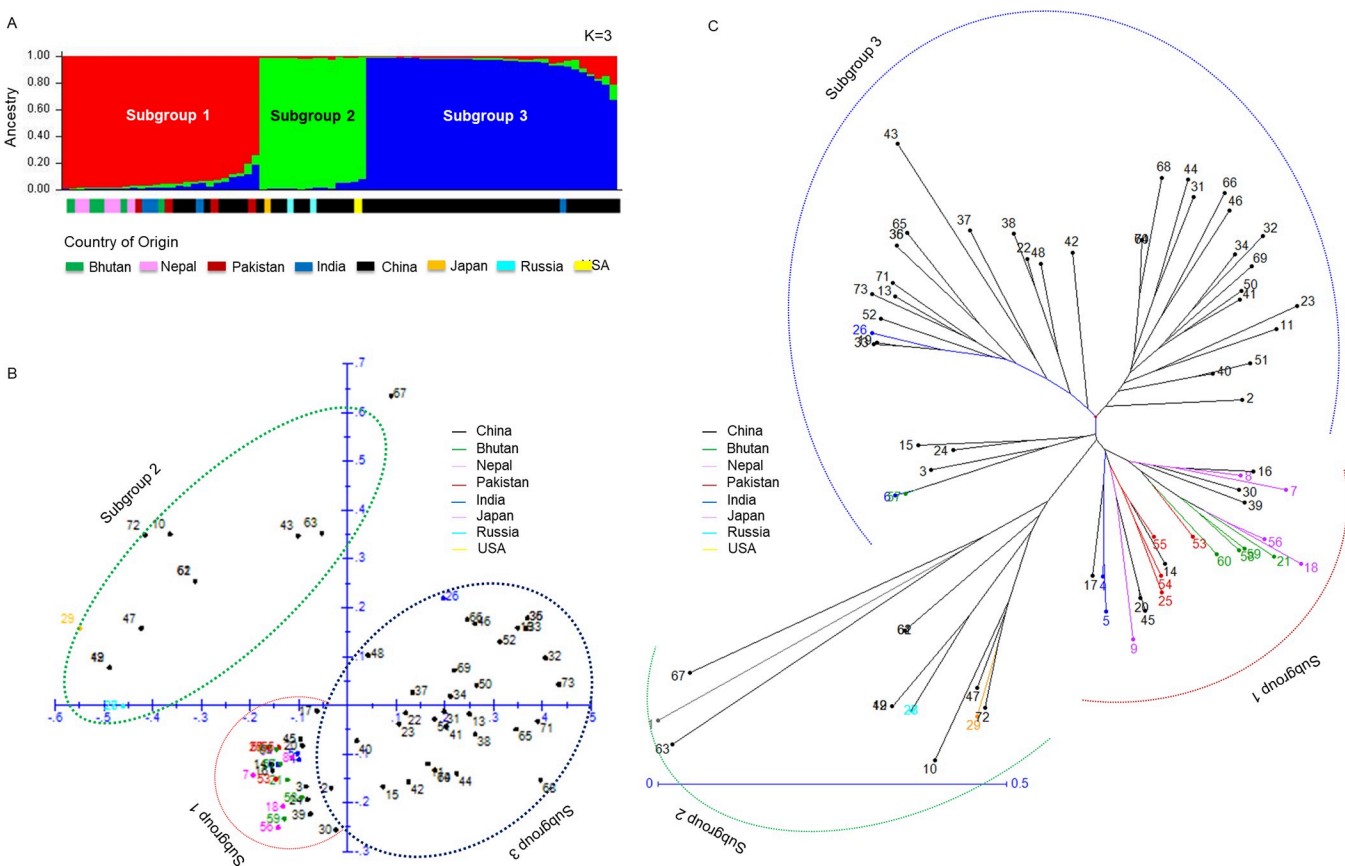

**Fig 8. Diversity analysis of tartary buckwheat accessions.** (A) STRUCTURE [63] results based on 50 InDels: 73 accessions were divided in 3 subpopulations. (B) Principal component analysis and STRUCUTURE classification were similar. Each color represents country of origin. (C) Phylogenetic analysis among 73 tartary buckwheat accessions using weighted neighbor-joining analysis in DARwin software [62]. The horizontal bar indicates distance based on the simple matching coefficient. Each subgroup represents a subpopulation based on STRUCTURE results.

(65.4%), while subgroup 2 was comprised of Chinese accessions (71.4%), and subgroup 3 consisted of Chinese accessions (97.0%). At the same time, the Chinese accessions in subgroup 1 and 3 showed a mixed composition of both red, green, and blue components. A PCA also provided similar results, with the accessions forming a tight cluster separated by collection region.

Weighted neighbor-joining (NJ tree) relationships with bin maps revealed 3 groups (G1, G2, and G3) that contained 40, 16, and 17 accessions, respectively (Fig 8). The accessions collected in Nepal and Bhutan demonstrated substructure within group 1, being the genotypes closest to the reference group. A different result revealed that two separate groups were clustered by using non-synonymous SNP variants (Fig 7). The color-coded branches supported the classification of the 73 accessions by the collection region (Fig 8). In the phylogenetic analysis, PCA and population structure indicated that the tartary buckwheat accessions segregated into different groups that reflected their geographical distribution. With the evidence relating the 50 InDels to environmental adaptability, these results are consistent with the segregation of the tartary buckwheat into different groups according to their geographical distribution. These results show the practicability of the barcode system, which can be used to identify genetic variation associated with phenotypic variation of functional relevance, as well as markers that can be used in crop breeding.

## Discussion

Tartary buckwheat is an important crop that is poised to be the target of many future breeding efforts. It is a crop species of interest for various reasons, including its higher levels of rutin compared to the common buckwheat [10,23,65]. For breeding commercial varieties of tartary buckwheat, the elite germplasm needs to be expanded by adding various genetic resources. This is a field that is still under-represented at the genetic level, despite its vast germplasm diversity and its importance in stress adapted resources [66–70]. Particularly, since these resources possess many distinct traits, we used 73 genetic resources of tartary buckwheat to create a framework for comparative analysis. These resources differ in seed morphology (Fig 1) and rutin content (Fig 2). These differences might result in distinct responses to UV-B, drought [69], and salt stress [67]. A recent publication of tartary buckwheat genomes provided the reference genome needed to study the molecular markers to distinguish between these genetic resources [48]. The narrow genetic diversity seen within a commercial variety has various causes, including the rigid quality required by farmers and processors [28,30]. Thus, molecular markers are imperative to understand the genetic differences of tartary buckwheat accessions, which could be used to broaden the genetic resources of commercial breeding programs. To address this, we made a user-friendly barcode system using 50 InDel-based genotypes of 73 accessions. Our barcode system could be used in genetic research, breeding programs, and efficient resource management systems for buckwheat.

High-throughput sequencing technologies [49] offer immense possibilities for generating a massive amount of sequencing data from hitherto uncharacterized genomes, which facilitates the introduction of InDels as genetic markers in non-model crops, like tartary buckwheat. Given that the existing reference genome for "Pinku 1" was from China [48], we constructed a new superscaffold-scale reference genome for a variety of "Daegwan 3–7" from the Republic of Korea by combining the sequencing strategies of PacBio and Illumina Hiseq2500. Using this integrated approach, we obtained an N50 scaffold size of 463,432 bps for the draft genome of "Daegwan 3–7" (Table 1). To explore the genomic variation of tartary buckwheat accessions, we performed whole-genome sequencing for 26 accessions across the reference genome ("Daegwan 3–7") (Table 2). Significantly, the depth (23 × – 27 ×) of sequence coverage indicates that it was well-suited for the sequence analysis of the 26 genetic resources. From these genome comparisons, 171,926 homogeneous InDels and 53,755 heterogeneous InDels were obtained with an InDel density of 2 for every 1 kb through pairwise comparison with the reference genome (Fig 3 and S5 Table). Among them, 50 polymorphic InDels were selected by gel electrophoresis, which PCR results were 98.2% matched with the predicted results (S6 and S7 Tables). Moreover, the 49 InDels were distributed widely in the whole genome of "Pinku 1." Considering that the TB8 locus was not found in "Pinku 1," the two reference genomes for "Pinku 1" and "Daegwan 3–7" can provide backbones and references for developing the pan-genome to reduce reference bias. We plan to continue improve the draft genome of tartary buckwheat cv. "Daegwan 3–7" for use in the pan-genome. Our results suggest that superscaffold-scale *de novo* draft genomes usually facilitate the introduction of InDels as genetic markers in non-model crops, like tartary buckwheat.

To efficiently identify tartary buckwheat accessions, we developed the barcode system using 50 InDels selected from 26 landrace genomes using a resequencing strategy. The 2D-barcode system (Figs 5 and 6) is a powerful tool to compare the genotypes of tartary buckwheat accessions [38–40]. The landrace genomes collected from six countries, contained a wealth of genetic diversity adapted to wide-ranging environmental conditions (Table 2). Through the application of 50 InDels to this collection, we found that both geographical and environmental factors have shaped the genetic diversity of tartary buckwheat landraces (Fig 7). In the 73

accessions, the population structure analyses using the DNA barcode data revealed their geographical distribution (Fig 8), which is geared for adaptation in the diverse topographic conditions of the agroecosystem [71–74]. In the phylogenetic analysis using non-synonymous SNPs, the 26 accessions showed a close relationship with those from Bhutan and Pakistan. (Fig 7). This observation is consistent with a previous study, which found that population genetic structure analysis using SSR data clustered one group that was mainly distributed in Nepal, Bhutan, and the Yunnan-Guizhou Plateau regions of China and the other group principally derived from the Loess Plateau regions, Hunan and Hubei of China and the USA [9]. These findings were consistent with the previously suggested method of seed exchange within a close geographic distance [5,8]. Altogether with SSR [9], AFLP [41], RAPD [42,43], and ISSR [44], the barcode system using InDels can complement current research and broaden our understanding of both phylogenetic signals and population-level variations [38–40]. These phylogenetic and structural analyses in the tartary buckwheat accessions will serve as a significant resource to gain more insight into the evolution of lineage-specific gene networks associated with the environmentally-specific developmental and adaptive traits of tartary buckwheat landraces.

According to their position in unigenes, the 50 InDels were divided into five types: coding region, 5′ untranslated region, 3′ untranslated region, splice region and intergenic region (S7 Table). The InDels may be biased toward changes in gene expression via cis-regulatory mutations [75]. The InDels of certain genes, e.g., *RPP4* [76], *UGT73B5* [77], *TPS4* [78], and *DREB3* [79] have been postulated to play a role in different responses to environmental stresses during growth and survival. With the evidence relating the 50 InDels to environmental adaptability, these results are consistent with the segregation of the tartary buckwheat into different groups according to their geographical distribution.

Despite self-fertilizing the tartary buckwheat, through 3 additional generations using the SSD method, around 10% heterogeneous InDels were detected in the genomic DNA from the 26 progenies (S3 Table). Previous studies have shown that farmers cultivated several tartary buckwheat landraces suitable for the multiple local microclimates and environmental conditions [5], which could drive the gene flow of landraces within a region [80]. In this study, we uncovered strong support for allogamy among landraces that contributes to heterogeneous InDels in tartary buckwheat, which might drive the genetic diversity of landraces within a region. Considering the effect of heterozygous InDel on the increase of single nucleotide mutation rate [75,81], it might be an important characteristic that allows tartary buckwheat to adjust to the diverse topographic conditions of the agroecosystem. These results suggest that the divergent gene expression of the InDel loci may affect its adaptation to environmental conditions.

We found that tartary buckwheat accessions could be distinguished by 50 InDels based on genotype according to location with the differing environment (Figs 7 and 8). Tartary buckwheat is a diverse crop in which separate domestication events happened in each gene pool followed by race and market class diversification that has resulted in different morphological characteristics in each commercial market class [82]. Combining multiple data types can compensate for missing or unreliable information in any single data type. Importantly, complete resource management is only likely to be discovered if the genotype, phenotype and chemotype of tartary buckwheat accessions are considered in an analysis. To make a user-friendly common platform for genotype, phenotype, and chemotype resources, we plan to incorporate genotypic data with that phenotype and chemotype (rutin content) data for tartary buckwheat accessions (S3 Fig). Our platform could be used in genetic research, breeding programs, and for efficient resource management systems for buckwheat. Increasing tartary buckwheat productivity will require adapting the crop to the agricultural environment of new locations. The

barcode system can assist tartary buckwheat breeders in identifying germplasm accessions and can be used as donor parents for breeding tartary buckwheat cultivars.

## Supporting information

**S1 Fig. The draft genome assembly process using *Fagopyrum tataricum* cv. "Daegwan 3–7".**
(TIF)

**S2 Fig. Agarose gels showing the separation of alleles of 50 InDels in 73 tartary buckwheat accessions.**
(PDF)

**S3 Fig. Application of barcode system to tartary buckwheat accession, HLB1003.**
(TIF)

**S1 Table. List of 73 tartary buckwheat accessions used for evaluating the selected InDels.**
(XLSX)

**S2 Table. A gradient UPLC method for the simultaneous determination of rutin and quercetin.**
(DOCX)

**S3 Table. Statistics of filtered raw data for evaluating Illumina paired read quality in 26 tartary buckwheat accessions.**
(DOCX)

**S4 Table. Number of InDels showing alternative genotype in 26 re-sequenced genomes.**
(DOCX)

**S5 Table. The 50 InDel markers developed to discriminate tartary buckwheat accessions.**
(XLSX)

**S6 Table. A diagram of PCR band patterns of 73 tartary buckwheat accessions obtained using 50 InDel markers.**
(XLSX)

**S7 Table. Statistics of the 50 InDels in tartary buckwheat accessions (n = 73).**
(DOCX)

## Acknowledgments

Sincere thanks to Misook Lee for the cultivation of tartary buckwheat resources, Juyoung Shim for the DNA extractions of samples, and Hyeongjin Shin for the technical assistance in the laboratory. The authors would like to thank Enago (www.enago.co.kr) for the English language review.

## Author Contributions

**Conceptualization:** Hwang-Bae Sohn, Su-Jeong Kim, Sunghoon Lee, Yul-Ho Kim.

**Data curation:** Hwang-Bae Sohn, Sin-Gi Park, Dong-Ha Oh, Yul-Ho Kim.

**Formal analysis:** Hwang-Bae Sohn, Sin-Gi Park, Dong-Ha Oh, Yul-Ho Kim.

**Funding acquisition:** Su-Jeong Kim, Yul-Ho Kim.

**Investigation:** Hwang-Bae Sohn, Su-Jeong Kim, Yul-Ho Kim.

**Methodology:** Hwang-Bae Sohn, Su-Jeong Kim, Yul-Ho Kim.

**Project administration:** Su-Jeong Kim, Yul-Ho Kim.

**Resources:** Su-Jeong Kim, Su-Young Hong, Jung Hwan Nam, Yul-Ho Kim.

**Supervision:** Yul-Ho Kim.

**Validation:** Hwang-Bae Sohn, Su-Jeong Kim, Yul-Ho Kim.

**Visualization:** Hwang-Bae Sohn, Su-Jeong Kim, Yul-Ho Kim.

**Writing – original draft:** Hwang-Bae Sohn, Hwa Yeun Nam.

**Writing – review & editing:** Hwang-Bae Sohn, Yul-Ho Kim.

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
