## [Decision Letter · Decision Letter 0]

5 Nov 2020

PONE-D-20-30493

Development of 50 InDel-based barcode system for genetic identification of tartary buckwheat resources

PLOS ONE

Dear Dr. Kim,

Thank you for submitting your manuscript to PLOS ONE. After careful consideration, we feel that it has merit but does not fully meet PLOS ONE’s publication criteria as it currently stands. Therefore, we invite you to submit a revised version of the manuscript that addresses the points raised during the review process.

We look forward to receiving your revised manuscript.

Kind regards,

Himanshu Sharma

Academic Editor

PLOS ONE

Additional Editor Comments:

Authors have done good work, for the genetic improvement in buckwheat, and the manuscript entitled "Development of 50 InDel-based barcode system for genetic identification of tartary buckwheat resources" has been extensively reviewed by the reviewers, They suggested revision of the manuscript, So the authors extensively revised the manuscript according to the suggestions of the reviewer.So there are chances of the improvement in the current manuscript.

Journal Requirements:

We note that one or more of the authors are employed by a commercial company: TherageneEtex Inc. and EONE-DIAGNOMICS Genome Center Co. Ltd.

2.1. Please provide an amended Funding Statement declaring this commercial affiliation, as well as a statement regarding the Role of Funders in your study. If the funding organization did not play a role in the study design, data collection and analysis, decision to publish, or preparation of the manuscript and only provided financial support in the form of authors' salaries and/or research materials, please review your statements relating to the author contributions, and ensure you have specifically and accurately indicated the role(s) that these authors had in your study. You can update author roles in the Author Contributions section of the online submission form.

3.2. Please also provide an updated Competing Interests Statement declaring this commercial affiliation along with any other relevant declarations relating to employment, consultancy, patents, products in development, or marketed products, etc.  

Reviewers' comments:

Reviewer's Responses to Questions

**Comments to the Author**

1. Is the manuscript technically sound, and do the data support the conclusions?

Reviewer #1: Yes

Reviewer #2: Yes

Reviewer #3: Partly

2. Has the statistical analysis been performed appropriately and rigorously? 

Reviewer #1: Yes

Reviewer #2: N/A

Reviewer #3: Yes

3. Have the authors made all data underlying the findings in their manuscript fully available?

Reviewer #1: Yes

Reviewer #2: Yes

Reviewer #3: Yes

4. Is the manuscript presented in an intelligible fashion and written in standard English?

Reviewer #1: No

Reviewer #2: No

Reviewer #3: Yes

5. Review Comments to the Author

Reviewer #1: Comments on the manuscript PONE-D-20-30493

The manuscript PONE-D-20-30493, entitled “Development of 50 InDel-based barcode system for genetic identification of tartary buckwheat resources”, developed 50 InDels to instruct a barcode system to increase the genetic diversity of tartary buckwheat. The results are interesting for identification and assessment of genetic diversity of tartary buckwheat resources. However, there are some problems in this manuscript. I suggest that this manuscript should make a major revision before being accepted for publication.

1. The language of this manuscript should be improved by a native English speaker, or a professional language editing service. There are a lot of grammar errors in this manuscript, such as Lines 94-97.

2. The 50 InDel-based barcode system has a bias for genetic identification of tartary buckwheat resources. The amount of 50 InDel-based genotypes of 73 accessions is not enough.

Reviewer #2: Dear Authors,

You have generated substantial information on developing INDEL barcode system in tartary buckwheat which can be improve the reliability of identifying genetic diversity among buckwheat germplasm. I found this study quite helpful in targeting future buckwheat breeding programs but have some minor concerns before formally accepting it from my end:

Abstract has been casually written in terms of language used, especially lines 41-45. Please modify accordingly.

What age plant material was used for measuring the rutin and quercetin contents? Mature….how many years?

What do you mean by ‘etc.’ in Fig. 2? I am sorry but you can’t use such terms especially describing a figure. Correct it in figure legend as well as in the figure itself. You can add country names as and what available.

In Fig. 3, replace the word ‘InDel’ by ‘InDels’ as you are dealing with a large no.

References need thorough rechecking as per journal standards. Maintain uniformity while citing edited books, chapters and/or proceedings.

Lastly, rectify the grammatical mistakes and sentence framing. Authors seemed to be in a hurry while compiling this analysis. It will be better if help of some native English speaker could be taken.

Regards

Reviewer #3: In this manuscript, the authors used combination of Pacbio long reads and Illumina short reads for draft genome assembly of tartary buckwheat cv. ‘Daegwan 3-7a and through whole-genome resequencing of 26 accessions had identified 171,926 homogeneous and 53,755 heterogeneous InDels. The technical approach chosen for this study is sound and the processing steps presented are commonly used for this kind of study.

However, there are few major concerns in the current manuscript, which authors should address

A. The authors write that they have represented a high quality draft genome assembly and annotation for tartary buckwheat. They do not provide the needed analysis results to backup this assertion, as they have not compared their assembly with previously available assemblies.

B. There is a need to include a large number of accessions representing any countries and indels for genetic identification of tartary buckwheat resources.

There are also some minor points need to be addressed

1. The material and methods section of the manuscript is not precise enough and does not contain all the processing steps presented in the result section. Such as the conditions for structure analysis should be detailed, with parameter settings.

2. The manuscript needs to be rechecked for grammatical errors and spelling mistakes.

3. Resolution of the all the figures should be checked to comply quality parameters of the journal.

4. In figure 5 A author must include a DNA base pair ladder.

5. In Figure 8 B there is a need to represent the differentiate accessions in their respective subgroups by encircling them.

6. PLOS authors have the option to publish the peer review history of their article (what does this mean?). If published, this will include your full peer review and any attached files.

Reviewer #1: No

Reviewer #2: No

Reviewer #3: **Yes: **Pradeep Singh

---

## [Author Response · Author response to Decision Letter 0]

4 Feb 2021

We greatly appreciate the reviewer’s efforts to carefully review our manuscript and the valuable suggestions offered, which were invaluable in helping us to make improvements. We have revised and modified our manuscript to clarify and strengthen its content, after taking into consideration all of the reviewers’ comments. Please find our responses to each of the reviewers’ comments in the uploaded file.

---

## [Decision Letter · Decision Letter 1]

23 Feb 2021

PONE-D-20-30493R1

Development of 50 InDel-based barcode system for genetic identification of tartary buckwheat resources

PLOS ONE

Dear Dr. Yul-Ho Kim

Thank you for submitting your manuscript to PLOS ONE. After careful consideration, we feel that it has merit but does not fully meet PLOS ONE’s publication criteria as it currently stands. Therefore, we invite you to submit a revised version of the manuscript that addresses the points raised during the review process.

We look forward to receiving your revised manuscript.

Kind regards,

Himanshu Sharma

Academic Editor

PLOS ONE

Journal Requirements:

Additional Editor Comments (if provided):

Based on the recommendations of the reviewer's authors had significantly revised the manuscript, and by analyzing the the revised manuscript the suggested that manuscript can be accepted for publication.

But my question is that authors have not deposit the sequencing data in Public repositories like NCBI or any other. Firstly authors submit data into NCBI and put SRA number or any ID, Because I don't get the details for the same in the manuscript and I am not satisfied with this, I only accept the manuscript if authors provide these details.

Reviewers' comments:

Reviewer's Responses to Questions

**Comments to the Author**

1. If the authors have adequately addressed your comments raised in a previous round of review and you feel that this manuscript is now acceptable for publication, you may indicate that here to bypass the “Comments to the Author” section, enter your conflict of interest statement in the “Confidential to Editor” section, and submit your "Accept" recommendation.

Reviewer #1: All comments have been addressed

Reviewer #2: All comments have been addressed

Reviewer #3: All comments have been addressed

2. Is the manuscript technically sound, and do the data support the conclusions?

Reviewer #1: Yes

Reviewer #2: Yes

Reviewer #3: Yes

3. Has the statistical analysis been performed appropriately and rigorously? 

Reviewer #1: Yes

Reviewer #2: Yes

Reviewer #3: Yes

4. Have the authors made all data underlying the findings in their manuscript fully available?

Reviewer #1: Yes

Reviewer #2: Yes

Reviewer #3: Yes

5. Is the manuscript presented in an intelligible fashion and written in standard English?

Reviewer #1: Yes

Reviewer #2: Yes

Reviewer #3: Yes

6. Review Comments to the Author

Reviewer #1: The authors have revised their manuscript according to the reviewers' comments. So I suggest that the manuscript should be accepted now.

Reviewer #2: All the concerned queries have been addressed satisfactorily. While language was a constraint initially, it has also been improved.

Reviewer #3: (No Response)

7. PLOS authors have the option to publish the peer review history of their article (what does this mean?). If published, this will include your full peer review and any attached files.

Reviewer #1: No

Reviewer #2: **Yes: **Dr. Nikhil Malhotra, ICAR-National Bureau of Plant Genetic Resources Regional Station, Shimla, India

Reviewer #3: **Yes: **Pradeep Singh

---

## [Author Response · Author response to Decision Letter 1]

22 Mar 2021

Dear Editor,

We greatly appreciate the reviewer’s efforts to carefully review our manuscript and the valuable suggestions offered, which were invaluable in helping us to make improvements. We have revised and modified our manuscript to clarify and strengthen its content, after taking into consideration all of the Editor’ comments. Please find our responses to each of the Editor’ comments below.

Please review your reference list to ensure that it is complete and correct. If you have cited papers that have been retracted. Please include the rationale for doing so in the manuscript text, or remove these references and replace them with relevant current references. Any changes to the reference list should be mentioned in the rebuttal letter that accompanies your revised manuscript. If you need to cite a retracted article, indicate the article’s retracted status in the References list and also include a citation and full reference for the retraction notice. 

The changed references are as follows. In addition, the changes have been indicated in the file (Revised Manuscript with Track Changes.docx).

Line 30 to 31:

We developed PCR-based and co-dominant insertion/deletion (InDel) markers to discriminate tartary buckwheat genetic resources.

Line 101 to 103:

In addition, Sohn et al. [38] reported that the InDel-based barcode system focuses on usability, and provided an efficient resource management system.

Line 130 to 133:

We grew an additional four generations of HLB1001 using the SSD method, which resulted in the selection of a progeny, “Daegwan 3-7” for a draft genome of tartary buckwheat. A total of 26 accessions were collected (China, India, Nepal, Bhutan, Pakistan, and USA).

Line 528 to 530:

Ohnishi O, Matsuoka Y. Search for the wild ancestor of buckwheat II. Taxonomy of Fagopyrum (Polygonaceae) species based on morphology, isozymes and cpDNA variability. Genes Genet Syst. 1996;71:383-390. doi: 10.1266/ggs.71.383 

Line 540 to 542:

Huang W, Jarvis D, Ahmed S, Long C. Tartary buckwheat genetic diversity in the Himalayas associated with farmer landrace diversity and low dietary dependence. Sustainability. 2017;9(10):1806. doi: 10.3390/su9101806

Line 543 to 545:

Fagopyrum esculentum Moench. X. Diffusion routes revealed by RAPD markers. Genes Genet Syst. 1996;71:211-218. doi: 10.1266/ggs.71.211 

Line 546 to 549:

Ohnishi O. Search for the wild ancestor of buckwheat III. The wild ancestor of cultivated common buckwheat, and of tatary buckwheat. Econ Bot. 1998;52(2):123-133. Available at http://www.jstor.com/stable/4256049.pdf. Accessed 18 September 2020

Line 553 to 556:

Hou S, Sun Z, Linghu B, Xu D, Wu B, Zhang B, et al. Genetic diversity of buckwheat cultivars (Fagopyrum tartaricum Gaertn.) assessed with SSR markers developed from genome survey sequences. Plant Mol Biol Report. 2016;34(1):233–241. doi: 10.1007/s11105-015-0907-5

Line 565 to 568:

Liu M, Ma Z, Zheng T, Sun W, Zhang Y, Jin W, et al. Insights into the correlation between physiological changes in and seed development of tartary buckwheat (Fagopyrum tataricum Gaertn.). BMC Genomics. 2018;19(1):648. doi: 10.1186/s12864-018-5036-8

Line 569 to 571:

Spengler R. Anthropogenic seed dispersal: Rethinking the origins of plant domestication. Trends Plant Sci. 2020;25(4):340-348. doi: 10.1016/j.tplants.2020.01.005

Line 582 to 584:

Fabjan N, Rode J, Košir IJ, Wang Z, Zhang Z, Kreft I. Tartary buckwheat (Fagopyrum tataricum Gaertn.) as a source of dietary rutin and quercitrin. J Agric Food Chem. 2003;51(22):6452-6455. doi: 10.1021/jf034543e

Line 604 to 606:

Wang Y, Campbell CG. Tartary buckwheat breeding (Fagopyrum tataricum L. Gaertn.) through hybridization with its rice-tartary type. Euphytica. 2007;156(3):399-405. doi: 10.1007/s10681-007-9389-3

Line 616 to 618:

Hyten DL, Song Q, Zhu Y, Choi IY, Nelson RL, Costa JM, et al. Impacts of genetic bottlenecks on soybean genome diversity. Proc Natl Acad Sci U S A. 2006;103(45):16666–16671. doi: 10.1073/pnas.0604379103

Line 619 to 621:

Bhandari HR, Bhanu AN, Srivastava K, Singh MN, Shreya, Hemantaranjan A. Assessment of genetic diversity in crop plants – an overview. Adv Plants Agric Res. 2017;7(3):279-286. doi: 10.15406/apar.2017.07.00255

Line 624 to 626:

Hughes AR, Inouye BD, Johnson MTJ, Underwood N, Vellend M. Ecological consequences of genetic diversity. Ecol Lett. 2008;11(6):609-623. doi: 10.1111/j.1461-0248.2008.01179.x

Line 630 to 632:

Väli Ü, Brandström M, Johansson M, Ellegren H. Insertion-deletion polymorphisms (indels) as genetic markers in natural populations. BMC Genet. 2008;9(1):8. doi: 10.1186/1471-2156-9-8

Line 639 to 642:

Meng HT, Zhang YD, Shen CM, Yuan GL, Yang CH, Jin R, et al. Genetic polymorphism analyses of 30 InDels in Chinese Xibe ethnic group and its population genetic differentiations with other groups. Scientific Reports. 2015;5(5):8260. doi: 10.1038/srep08260

Line 639 to 642:

Tsuji K, Ohnishi O. Phylogenetic position of east Tibetan natural populations in tartary buckwheat (Fagopyrum tataricum Gaertn.) revealed by RAPD analyses. Genet Resour Crop Evol. 2001;48(1):63–67. doi: 10.1023/A:1011286326401

Line 639 to 642:

Zhang A, Zhao L. Molecular characterization of genetic diversity of underutilized crops; Buckwheat as an example. Acta Hortic. 2013;979:407-419. doi: 10.17660/ActaHortic.2013.979.44

Line 718 to 720:

Liu K, Muse SV. PowerMarker: an integrated analysis environment for genetic marker analysis. Bioinformatics. 2005;21(9):2128-2129. doi: 10.1093/bioinformatics/bti282

Line 721 to 722:

Perrier X, Jacquemoud-Collet JP. DARwin software. 2006. Available at http://darwin.cirad.fr/

Line 728 to 731:

Jiang P, Burczynski F, Campbell C, Pierce G, Austria JA, Briggs CJ. Rutin and flavonoid contents in three buckwheat species Fagopyrum esculentum, F. tataricum, and F. homotropicum and their protective effects against peroxidation. Food Res Int. 2007;40(3):356-364. doi: 10.1016/j.foodres.2006.10.009

Line 741 to 743: 

Germ M, Brenznik B, Dolinar N, Kreft I, Gaberščik A. The combined effect of water limitation and UV-B radiation on common and tartary buckwheat. Cereal Res Commun. 2013;41(1):97-105. doi: 10.1556/CRC.2012.031

Line 744 to 747: 

Pandey V, Niranjan A, Atri N, Chandrashekhar K, Mishra MK, Trivedi PK, Misra P. WsSGTL1 gene from Withania somnifera, modulates glycosylation profile, antioxidant system and confers biotic and salt stress tolerance in transgenic tobacco. Planta. 2014;239(6):1217-1231. doi: 10.1007/s00425-014-2046-x

Line 762 to 764: 

Tian D, Wang Q, Zhang P, Araki H, Yang S, Kreitman M, et al. Single-nucleotide mutation rate increases close to insertion/deletions in eukaryotes. Nature. 2008;455:105-108. doi: 10.1038/nature07175

Line 765 to 768: 

van der Biezen EA, Freddie CT, Kahn K, Parker JE, Jones JDG. Arabidopsis RPP4 is a member of the RPP5 multigene family of TIR-NB-LRR genes and confers downy mildew resistance through multiple signaling components. Plant J. 2002;29(4):439-451. doi; 10.1046/j.0960-7412.2001.01229.x

Line 769 to 772: 

Langlois-Meurinne M, Gachon CM, Saindrenan P. Pathogen-responsive expression of glycosyltransferase genes UGT73B3 and UGT73B5 is necessary for resistance to Psudomonas syringae pv tomato in Arabidopsis. Plant Physiol. 2005;139(4):1890-1901. doi: 10.1104/pp.105.067223

Line 773 to 776: 

Attaran E, Rostás M, Zeier J. Pseudomonas syringae elicits emission of the terpenoid (E,E)-4,8,12-trimethnyl-1,3,7,11-tridecatetraene in Arabidopsis leaves via jasmonate signaling and expression of the terpene synthase TPS4. Mol Plant Microbe Interact. 2008;21(11):1482-1497. doi: 10.1094/MPMI-21-11-1482 

Line 777 to 779: 

Islam MS, Wang MH. Expression of dehydration responsive element-binding protein-3 (DREB3) under different abiotic stresses in tomato. BMB Rep. 2009;42(9):611-616. doi: 10.5483/bmbrep.2009.42.9.611 

Line 780 to 782: 

Bellucci E, Bitocchi E, Rau D, Nanni L, Ferradini N, Giardini A, et al. Population structure of barley landrace populations and gene-flow with modern varieties. PLoS ONE. 2013;8(12):e83891. doi: 10.1371/journal.pone.0083891 

Line 783 to 785: 

Hollister JD, Ross-Ibara J, Gaut BS. Indel-associated mutation rate varies with mating system in flowering plants. Mol Biol Evol. 2010;27(2):409-416. doi: 10.1093/molbev/msp249 

Line 786 to 787: 

Leishman MR, Westoby M, Jurado E. Correlates of seed size variation: a comparison among five temperate floras. J Ecol. 1995;83(3):517-529. doi: 10.2307/2261604

My question is that authors have not deposit the sequencing data in Public repositories like NCBI or any other. Firstly authors submit data into NCBI and put SRA number or any ID. Because I don’t get the details for the same in the manuscript and I am not satisfied with this, I only accept the manuscript if authors provide these details.

I agree with your opinion. But, the draft genome has limitations. First, the draft genome of tartary buckwheat cv. “Daegwan 3-7” showed 43,771 gene models compared to 30,386 in “PinKu1.” Second, “Daegwan3-7” matched 80% of showed 80% gene matches compared to 89% in “Pinku1” out of 1,375 BUSCOs. Thus, we’re currently working with Professor Dong-Ha Oh of Louisiana State University to improve gene models and BUSCOs of the draft genome. In addition, we will try to present de novo assembly of the 26 genomes by using the improved gene models and BUSCOs of the draft genome. It would be possible to perform pan-genome analysis of the 26 genomes, enabling identification of the core and expanded subgenomes of tartary buckwheat. 

In the future, we will deposit the improved sequencing data in public repositories. These results will lead to the discovery of a larger number of genome-wide markers which will help researchers to genetically characterize the tartary buckwheat genome at high resolution. Furthermore, genome sequences, as well as orthologous genes and gene families identified as uniquely in tartary buckwheat through comparative analysis, provide an essential resource for studying the adaptation, specialized metabolism, and the genomic basis of the strikingly high rutin content in this interesting crop species.

It is clear that the 50-InDel based barcode system should be efficient, fast and low-cost approach for the genetic identification of tartary buckwheat accessions. We believe that these modifications have strengthened the manuscript and hope that the revised manuscript is suitable for publication in PLOS ONE.

---

## [Editor Report · Decision Letter 2]

14 Apr 2021

Development of 50 InDel-based barcode system for genetic identification of tartary buckwheat resources

PONE-D-20-30493R2

Dear Dr. Kim,

We’re pleased to inform you that your manuscript has been judged scientifically suitable for publication and will be formally accepted for publication once it meets all outstanding technical requirements.

Kind regards,

Himanshu Sharma

Academic Editor

PLOS ONE

Additional Editor Comments (optional):

Authors have addressed all the queries raised by reviewer and Academic Editor and putting one note that in future when this genome sequence is annotated and completed they must submit the sequence in public repositories. Now the manuscript is looking good, but there are always possibilities of correcting the mistakes which authors will take care at the time of proofread.
---

## [Editor Report · Acceptance letter]

7 May 2021

PONE-D-20-30493R2 

Development of 50 InDel-based barcode system for genetic identification of tartary buckwheat resources 

Dear Dr. Kim:

I'm pleased to inform you that your manuscript has been deemed suitable for publication in PLOS ONE. Congratulations! Your manuscript is now with our production department. 

Kind regards, 

on behalf of

Dr. Himanshu Sharma 

Academic Editor

PLOS ONE